# Assessing the Roles of Terrestrial Stilling and Solar Dimming in Land Surface Drying/Wetting across China

**Kai Duan [1,2,3,*], Jiali Guo [4,5,*], Tiesong Hu [2], Xianxun Wang [6] and Yadong Mei [2]**

1   School of Civil Engineering, Sun Yat-Sen University, Guangzhou 510275, China
2   Laboratory of Water Resources and Hydropower Engineering Science, Wuhan University, Wuhan 430070, China; tshu@whu.edu.cn (T.H.); ydmei@whu.edu.cn (Y.M.)
3   Southern Marine Science and Engineering Guangdong Laboratory (Zhuhai), Zhuhai 524043, China
4   College of Hydraulic & Environmental Engineering, China Three Gorges University, Yichang 443002, China
5   Engineering Research Center of Eco-environment in Three Gorges Reservoir Region, Ministry of Education, China Three Gorges University, Yichang 443002, China
6   College of Resources and Environment, Yangtze University, Wuhan 434025, China; xianxunwang@gmail.com
*   Correspondence: duank6@mail.sysu.edu.cn (K.D.); jiali.guo@ctgu.edu.cn (J.G.)

**Abstract:** Decreases in wind speed (i.e., terrestrial stilling) and radiation (i.e., solar dimming) have been identified as important causes of aridity change both globally and regionally. To understand how their roles have varied across different natural and socioeconomic circumstances in China, this study presents a nationwide attribution analysis of land surface drying/wetting across the ten first-level river basins. The results suggest that consistent warming and reductions in relative humidity have significantly enhanced atmospheric evaporative demand and driven the land surface to become drier over the past six decades. However, the widespread terrestrial stilling and solar dimming have largely offset such trends by suppressing evaporation. While spatially varying changes in precipitation were the most influential driver of aridity change over half of the 713 used climate sites, decreasing wind speed and radiation were identified as the dominant cause of wetting at 15% and 13% of the sites, respectively. The impacts of terrestrial stilling and solar dimming were generally more prominent in the north (e.g., the Liao River, Songhuajiang, Hai River, and Huai River basins) and south (e.g., the Southeast, Pearl River, and Yangtze River basins) respectively, which could be associated with the weakening monsoon and intensified anthropogenic disturbances such as ecological restoration, urbanization, and air pollution. We conclude that more attention needs to be paid to the independent and combined climatological impacts of global- and regional-level human activities to develop proactive adaptation strategies of water and land management.

**Keywords:** aridity; terrestrial stilling; solar dimming; attribution analysis; China

## 1. Introduction

There is evidence from both observation and simulation studies that climate change causes land surface drying in many regions across the world [1–3] which leads to depletion in water availability [4], larger irrigation water demand [5], more frequent wildfire [6], and land degradation and desertification [7]. Previous studies usually interpreted dryness from the perspective of either aridity (i.e., the long-term background state of moisture availability in ecosystems) [2,8] or drought (i.e., temporal anomalies of wet/dry condition) [1,9,10]. Changes in dryness/wetness status are usually measured by changes in precipitation, potential evapotranspiration (PET), and related water-balance components. While precipitation represents the total water supply for land surfaces, PET indicates the

maximum water exchange from the land to the atmosphere (i.e., atmospheric evaporative demand), which is determined by the status of ambient air and energy supplies. Although global average precipitation increases as the earth warms, a drier future is expected to occur as the warming climate increases the evaporative demand of the atmosphere to a degree that cannot be fully compensated by the increase in precipitation [11].

It has been well established that air temperature, solar radiation, air humidity, and wind speed are the key climatological parameters for assessing the evaporation process [12–14]. Many climate change studies only considered the effects of precipitation variability and rising temperature [15], i.e., modeling terrestrial water balance or evaluating dryness on the basis of temperature-based PET methods such as Thornthwaite [16] and Hamon [17]. These methods have been commonly used because they only require temperature data as input and are well validated for historical periods. Essentially, the physical base of these empirical equations is the correlation between temperature and other radiative and aerodynamic variables that control the evaporation and transpiration fluxes. Nevertheless, such correlation may not remain valid in a rapidly changing environment, and failure to explicitly account for changes in humidity, radiation, and wind could cause fundamental biases. Recent studies suggested that the bias in temperature-based methods could be amplified in a warming future, and thus lead to the overestimation of PET and the severity of drying [18,19]. For example, Duan et al. [15] investigated the independent effects of five major climatic variables on future runoff over the United States, and found that the increasing air humidity would largely offset the additional evaporative demand caused by warming and lead to a wetter future in the eastern United States.

On-site observations have also shown a decline in the aerodynamic or radiative components of PET over the globe, which are usually referred to as global terrestrial stilling [20] and global dimming [21], respectively. Such trends of terrestrial stilling and solar dimming can counterbalance the warming-induced increases in PET and alter regional aridity along with changes in precipitation. However, the timing and magnitude of the changes in different climatic variables, as well as their independent and combined effects on terrestrial water balance, vary greatly both temporally and spatially across the world [20,22] and have led to different drying or wetting characteristics at regional scales [23,24].

China covers a wide range of climatic, geophysical, and ecological circumstances. Previous studies have reported the spatiotemporal variations in PET and the associated drying/wetting characteristics in China from different perspectives [10,24,25]. The roles of specific climatic variables have also been investigated for various regions of China. For example, Xu et al. [26] found that decreasing solar radiation and wind speed caused reduction in both pan evaporation and PET in the Yangtze River basin in 1960–2000; Zhang et al. [27] discussed historical variations in reference to evapotranspiration and the influences of changing radiation and humidity caused by human activities; Liu et al. [28] investigated the causes of the "pan evaporation paradox" [29] in China. However, the relative contributions of changing climatic variables to aridity have not been rigorously quantified and compared across climatic regimes and river basins. Therefore, we aim to expand upon previous studies by providing an assessment of aridity variations and the key climatic driving factors using up-to-date on-site observations in China. Specifically, the goal of this study is: (1) to investigate the drying and wetting paradigms over China in recent decades; (2) to quantify the direct and compound effects of changing precipitation, temperature, wind, humidity, and solar radiation on aridity; and (3) to explore the linkages between the changing climate and large-scale anthropogenic disturbances and the implications for the hydrological cycle across the major river basins.

## 2. Methods

### 2.1. Climate Data

Daily historical climate records from 1951 to 2016 at 824 climate stations were obtained from the China Meteorological Administration (CMA). The up-to-date "China surface climate daily dataset

V3.0" (http://data.cma.cn/data) includes daily precipitation, maximum and minimum air temperature, wind speed, air pressure, relative humidity, and sunshine duration. The records from the 1950s were discarded for containing a large portion of missing values. The climate stations were distributed across the ten first-level basins in China, including the Songhuajiang (1), Liao River (2), Hai River (3), Yellow River (4), Huai River (5), Yangtze River (6), Southeast (7), Pearl River (8), Southwest (9), and Northwest (10) basins. The density of climate stations was generally in line with population density, and there were over 30 sites within each basin to represent the regional characteristics (Table 1). We excluded 98 out of the total available 824 sites for containing consecutive gaps longer than 365 days, and another 13 sites located outside of the 10 basins. Daily records from January 1961 to December 2016 at 713 sites (Figure 1) were eventually used for this analysis. Among the selected 713 stations, there were 22 sites containing more than 1% missing days in the precipitation data, and 174 sites containing gaps longer than one week. The durations of gaps in the records of temperature, humidity, wind, and sunshine duration were mostly only one day. The daily missing values were filled as follows. First, small gaps were filled by the averages of the two nearest days; then, consecutive gaps longer than one week were filled by the multi-decadal averages of the same days in other years to avoid affecting the long-term trends of climate series.

**Table 1.** Categories of aridity.

| Category | Aridity Index |
|---|---|
| Hyper-arid | <0.05 |
| Arid | 0.05 to 0.2 |
| Semi-arid | 0.2 to 0.5 |
| Dry sub-humid | 0.5 to 0.65 |
| Sub-humid | 0.65 to 0.75 |
| Humid | >0.75 |

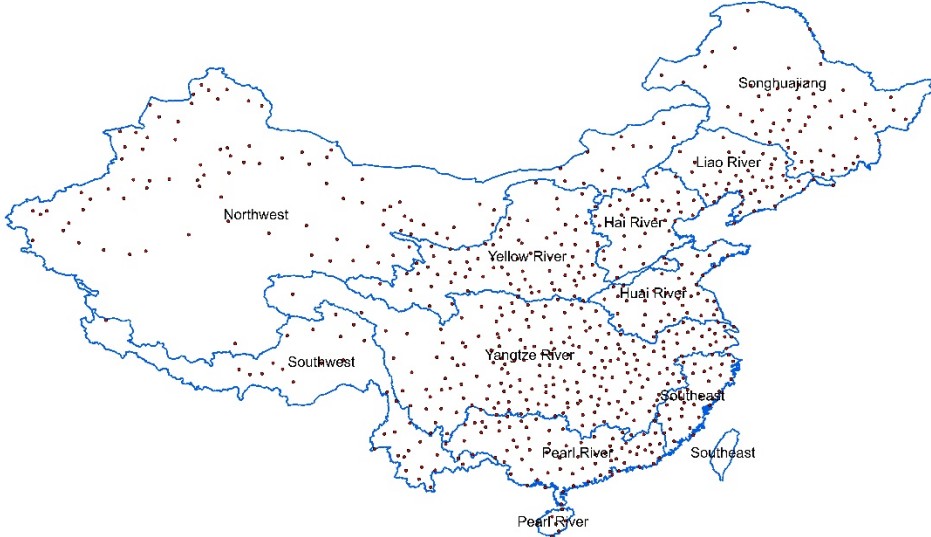

**Figure 1.** Distribution of the 713 used climate stations in the ten first-level basins in China. The ten basins include Songhuajiang (1), Liao River (2), Hai River (3), Yellow River (4), Huai River (5), Yangtze River (6), Southeast rivers (7), Pearl River (8), Southwest rivers (9), and Northwest rivers (10).

Radiation fluxes have not been measured as widely as other variables in China. We used the records of sunshine duration to estimate solar radiation, as:

$$Rs = \left(as + bs \cdot \frac{S}{N}\right) \cdot Ra \qquad (1)$$

where $S$ is daily sunshine duration (h); $N$ is the maximum possible duration of sunshine in a day (h); $Ra$ is daily extraterrestrial radiation (MJ m$^{-2}$ d$^{-1}$) that can be calculated by the solar constant, the latitude of the station, and the Julian date of the year [12]. The empirical coefficients $as$ and $bs$ were obtained from previous studies [8,25]: $as$ ranged from 0.12 to 0.29 and $bs$ ranged from 0.45 to 0.73, where these were calibrated and validated based on radiation observations from 116 stations across the country. The optimized radiation model using these $as$ and $bs$ values achieved a mean relative error of 6.5%, and has been proved useful for reflecting the impacts of regional climatic and geographic characteristics on radiation balance [25].

## 2.2. Dryness Indicator

The Aridity Index (AI), defined as the ratio of annual precipitation (P) to annual PET, was used to quantify long-term land dryness [30]. Drylands can be divided into hyper-arid (AI < 0.05), arid (0.05 ≤ AI < 0.2), semi-arid (0.2 ≤ AI < 0.5), and dry subhumid (0.5 ≤ AI < 0.65), while sub-humid and humid lands are regions with 0.65 ≤ AI < 0.75 and AI > 0.75 [31] (Table 1). PET was calculated at daily scale using the Penman–Monteith equation suggested by the Food and Agricultural Organization of the United Nations (FAO) [12], as:

$$\text{PET} = \frac{0.408\Delta(R_n - G) + \gamma \frac{900}{T+273} u_2 (e_s - e_a)}{\Delta + \gamma(1 + 0.34u_2)} \tag{2}$$

where $\Delta$ is slope of vapor pressure curve (kPa °C$^{-1}$); $R_n$ is net radiation (MJ m$^{-2}$ d$^{-1}$); $G$ is soil heat flux density (MJ m$^{-2}$ d$^{-1}$); $\gamma$ is the psychrometric constant (kPa °C$^{-1}$); $T$ is mean daily temperature (°C); $e_s - e_a$ is saturation vapor pressure deficit (kPa); $u_2$ is wind speed at 2 m height (m/s), which is converted from the CMA wind speed observed at 10 m height as:

$$u_2 = \frac{4.87u_{10}}{\log(67.8 \cdot Z - 5.42)} \tag{3}$$

where $Z$ is the elevation above sea level of the climate station (m). Compared with temperature-based and radiation-based PET methods, the Penman–Monteith PET is derived from physical principles and is considered the most reliable PET approach where sufficient meteorological data exist [13,19]. The FAO PET represents the evapotranspiration rate of a hypothetical well-watered grass reference crop with a height of 0.12 m, a surface resistance of 70 s m$^{-1}$, and an albedo of 0.23. Therefore, it is suitable for comparing the individual effects of surface temperature, air humidity, wind, and solar radiation across various climate and landscape conditions.

## 2.3. Attribution Analysis

The changes in AI are attributed to the impacts of changing P and PET as:

$$\frac{dAI}{dt} = \frac{1}{\text{PET}} \cdot \frac{dP}{dt} - \frac{P}{\text{PET}^2} \cdot \frac{dPET}{dt} + \frac{P}{\text{PET}^3} \cdot \left(\frac{dPET}{dt}\right)^2 \tag{4}$$

where $\frac{dAI}{dt}$ is the change in the ratio of P to PET; $\frac{1}{\text{PET}} \cdot \frac{dP}{dt}$ is the independent effect of P change to AI change; the sum of the second and third terms on the right side of the equation is the independent effect of PET change [2]. Then, the PET effect is further attributed to changing temperature (including max and min temperature, i.e., Tmax and Tmin), relative humidity (Rh), wind speed (Ws), and solar radiation (Rs) by a differentiating equation [28,32], as:

$$\begin{aligned} \frac{d\text{PET}}{dt} &= E(\text{Tmax}) + E(\text{Tmin}) + E(\text{Rh}) + E(\text{Ws}) + E(\text{Rs}) \\ &\approx \frac{\partial \text{PET}}{\partial \text{Tmax}} \cdot \frac{d\text{Tmax}}{dt} + \frac{\partial \text{PET}}{\partial \text{Tmin}} \cdot \frac{d\text{Tmin}}{dt} + \frac{\partial \text{PET}}{\partial \text{Rh}} \cdot \frac{d\text{Rh}}{dt} + \frac{\partial \text{PET}}{\partial \text{Ws}} \cdot \frac{d\text{Ws}}{dt} + \frac{\partial \text{PET}}{\partial \text{Rs}} \cdot \frac{d\text{Rs}}{dt} \end{aligned} \tag{5}$$

Given that the changing climatic variables may cause either positive or negative effects on AI, their contributions (%) are quantified by the relative weights, as:

$$C(i) = 100 \times \frac{|E(i)|}{\sum_{i=1}^{N} |E(i)|} \tag{6}$$

We here specifically focus on quantifying the climatic drivers of aridity change from 1961–1988 to 1989–2016. These two time periods were chosen for two reasons. First, the time series of climatic variables show different statistical variations (e.g., changing points, trends) across basins and sites. These two time periods, which each cover nearly 30 years, are the longest valid observations available at the national scale to provide a consistent comparison of the long-term climate change in different regions. Second, large-scale industrialization and urbanization in China sped up after the late 1980s, and in the meantime the environmental impacts of rapid socioeconomic developments have become more evident [33–35]. The nominal gross domestic product had grown eightfold (from 50 to 408 billion) from 1961 to 1988, yet grew twenty-five-fold (from 456 to 11,195 billion) from 1989 to 2016 (http://data.stats.gov.cn/english/). A rigorous quantification of aridity changes and the contributions of individual climatic drivers during these two time periods can provide useful information for understanding the climatological and hydrological impacts of industrialization and urbanization in China.

## 3. Results

### 3.1. Observed Long-Term Trends in Climate

The multi-decade average aridity in the time period of 1961–2016 (Table 2) suggests that the Southeast (AI = 1.89), Pearl River (1.78), Yangtze River (1.58), Huai River (1.05), and Southwest (1.01) basins were the most humid regions, where P exceeded the commonly used threshold value of 800 mm yr$^{-1}$ and Rh exceeded 70%. In northeastern China, Liao River and Songhuajiang were categorized as "humid" (0.81) and "sub-humid" (0.73), respectively. The Hai River (0.62) and Yellow River (0.55) basins were identified as "arid", or dryland. The Northwest basin was the driest region with the lowest P (183 mm yr$^{-1}$) and AI (0.17), where low Rh (49%) and high Rs (16 MJ m$^{-2}$ d$^{-1}$) had led to the highest basin-average PET of up to 1086 mm yr$^{-1}$.

We first used the Mann–Kendall test [36,37] to detect the long-term monotonic trends in the time series of aridity and climatic variables from 1961 to 2016 (Figure 2). Significant increasing trends in P at the 5% significance level were found at 11% of the sites, mostly in the Northwest, while significant decreases (8% of the sites) occurred across the Southwest, Pearl, Yangtze, and Yellow River basins. Meanwhile, increases in both Tmax and Tmin occurred at nearly all of the sites (>98%), and significant trends were detected at 84% and 95% of them. Widespread decreases were also found in the series of Rh, Ws, and Rs, with significant decreasing trends accounting for 45%, 69%, and 67% of the sites, respectively. In spite of the warming trend all over the country, the decrease in PET still covered as many as 68% of the sites (significant at 37%) due to the decreasing Ws and Rs. The spatial distributions of the trends of AI were generally similar to that of P, yet more increases (59%) than for P (48%) could be found as a result of the widespread decreases in PET. Note that most of these trends of AI (>80%) were not significant at the 5% significance level, reflecting the strong temporal variability in regional P and PET. Significant increasing (mainly in the Northwest, Songhuajiang, lower Yangtze, and Southeast) and decreasing (mostly in the Southwest, Yellow, upper Yangtze, and upper Pearl) trends of AI accounted for 11% and 6% of the sites, respectively.

**Table 2.** Mean annual precipitation (P), maximum temperature (Tmax), minimum temperature (Tmin), relative humidity (Rh), wind speed (Ws), radiation (Rs), potential evapotranspiration (PET), and aridity (AI) in 1961–2016 in the ten first-level basins in China.

| ID | Basin | Sites | Area ($10^4$ km$^2$) | P (mm) | Tmax (°C) | Tmin (°C) | Rh (%) | Ws (m/s) | Rs (MJ m$^{-2}$ d$^{-1}$) | PET (mm) | AI |
|---|---|---|---|---|---|---|---|---|---|---|---|
| 1 | Songhuajiang | 60 | 90 | 573 | 9.4 | −2.7 | 65.0 | 3.0 | 13.3 | 783 | 0.73 |
| 2 | Liao River | 45 | 35 | 709 | 13.7 | 2.3 | 62.0 | 2.8 | 13.4 | 873 | 0.81 |
| 3 | Hai River | 36 | 32 | 596 | 16.4 | 4.8 | 59.6 | 2.5 | 14.0 | 956 | 0.62 |
| 4 | Yellow River | 78 | 79 | 517 | 15.0 | 2.5 | 59.3 | 2.2 | 14.6 | 936 | 0.55 |
| 5 | Huai River | 43 | 33 | 981 | 19.4 | 10.0 | 71.5 | 2.7 | 12.9 | 930 | 1.05 |
| 6 | Yangtze River | 201 | 181 | 1369 | 20.5 | 11.9 | 75.9 | 1.9 | 11.6 | 866 | 1.58 |
| 7 | Southeast | 33 | 24 | 1766 | 23.3 | 14.6 | 78.5 | 1.8 | 12.4 | 933 | 1.89 |
| 8 | Pearl River | 84 | 58 | 1802 | 25.2 | 17.2 | 78.5 | 1.9 | 12.7 | 1011 | 1.78 |
| 9 | Southwest | 32 | 85 | 1035 | 19.3 | 6.5 | 64.2 | 1.6 | 16.1 | 1023 | 1.01 |
| 10 | Northwest | 101 | 337 | 183 | 13.9 | 0.1 | 49.2 | 2.7 | 16.0 | 1086 | 0.17 |

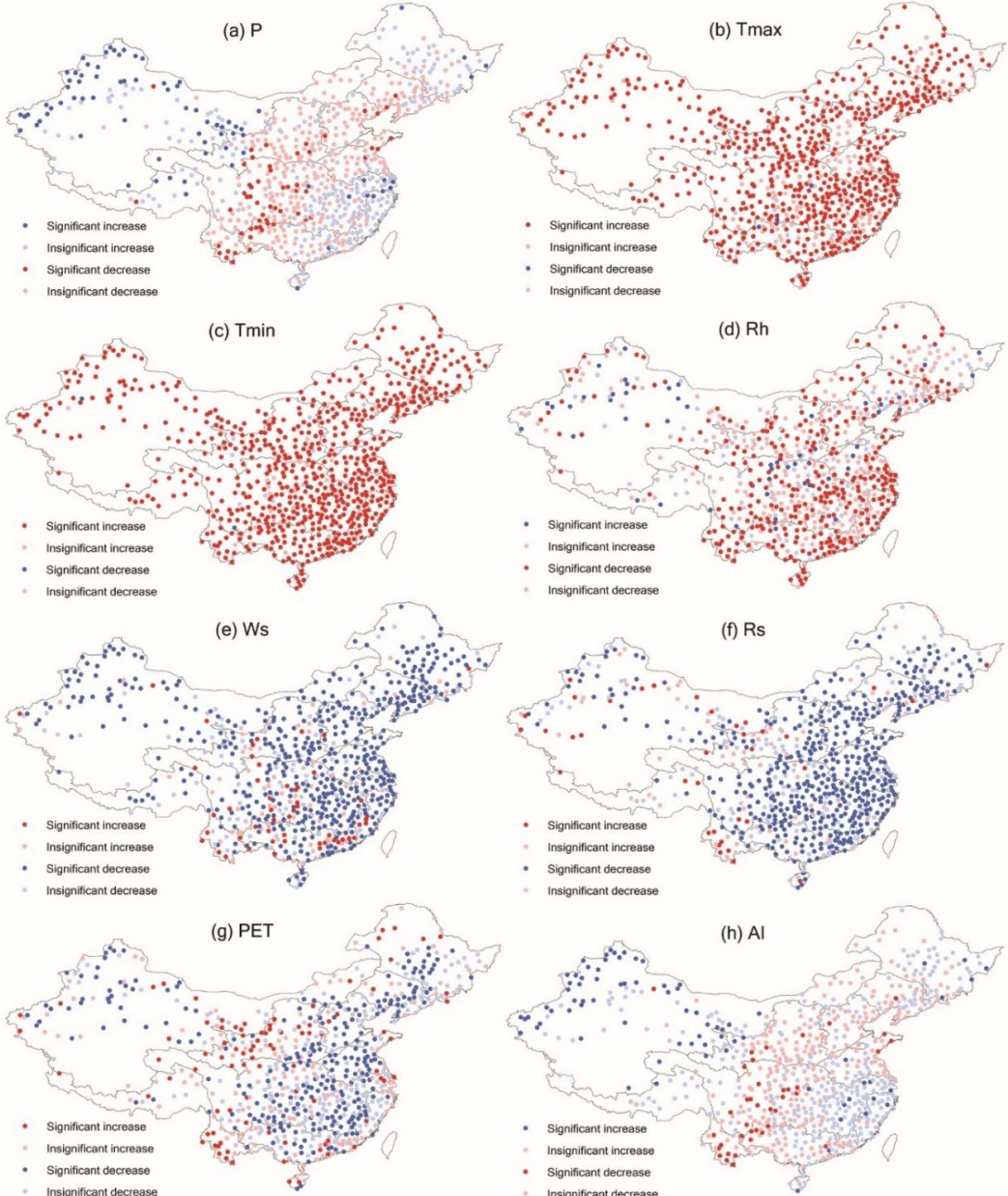

**Figure 2.** Trends of annual precipitation (P), maximum temperature (Tmax), minimum temperature (Tmin), relative humidity (Rh), wind speed (Ws), radiation (Rs), potential evapotranspiration (PET), and aridity index (AI) from 1961 to 2016 across the 713 climate stations. The trends were detected using the Mann–Kendall test at the significance level of 5%. The changes in climatic variables that drive AI to increase (i.e., increasing P and Rh; decreasing Tmax, Tmin, Ws, Rs, and PET) and decrease (i.e., decreasing P and Rh; increasing Tmax, Tmin, Ws, Rs, and PET) are denoted by blue and red dots, respectively.

*3.2. Drying/Wetting Patterns*

The changes in multi-decadal average climate from 1961–1988 to 1989–2016 were calculated to provide a quantitative evaluation of the extent and intensity of drying/wetting. Both the proportions of drying/wetting sites and the averaged aridity change within each basin (Figure 3) suggest that the Liao River, Hai River, Yellow River, Pearl River, and Southwest basins have been generally drying, while a wetting signal can be found in the Songhuajiang, Huai River, Yangtze River, Southeast, and Northwest basins.

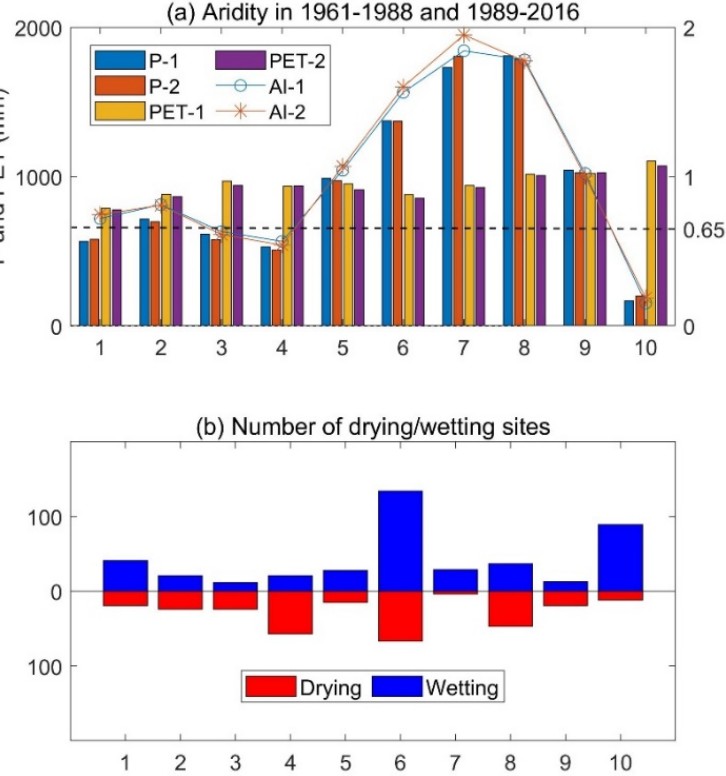

**Figure 3.** (**a**) Average precipitation (P), potential evapotranspiration (PET), and aridity index (AI) in 1961–1988 and 1989–2016 in the ten basins in China. (**b**) Number of sites getting drier or wetter from 1961–1988 to 1989–2016 within each basin. The first (1961–1988) and second time periods (1989–2016) are denoted by "-1" and "-2" in Figure 2a. On the x-axis, 1~10 represent the basins of Songhuajiang (1), Liao River (2), Hai River (3), Yellow River (4), Huai River (5), Yangtze River (6), Southeast (7), Pearl River (8), Southwest (9), and Northwest (10), respectively.

The counterpart results at the seasonal scale (Figure 4) show that climate change and the associated changes in aridity did not occur homogeneously within each year. In spring, it became more humid in the west (i.e., Southwest and Northwest) and northeast (i.e., Songhuajiang, Liao River, Hai River), but more arid in the south of the Huai River basin (i.e., Huai River, Yangtze River, Southeast, and Pearl River). Drying trends occurred widely in the Southwest and across northern China in summer, including the Songhuajiang, Liao River, Hai River, and Yellow River basins, while wetting trends were found in the south from the Huai River to the Pearl River basins. Similar spatial patterns can be found in autumn compared to the trends in spring, except that there were more drying sites in the northeast and the Southwest basin. Wetting signals were detected in winter at over 95% of all the sites.

The spatial distributions of drying and wetting signals suggest that the frequently discussed "dry gets drier, wet gets wetter" paradigm [38] only holds true at 51% of the 713 sites, including 78 out of the 220 (35%) dry sites (AI < 0.65) getting drier and 283 out of the 493 (57%) wet sites (AI > 0.65) getting wetter. The changes in aridity were caused by changes in precipitation and other climatic variables through their impacts on PET. We here categorize the aridity change across the 713 sites into four types, including water-driven drying/wetting where precipitation is identified as the largest contributor, and energy-driven drying/wetting where wind, radiation, or humidity is identified as the largest contributor (Figures 5 and 6). At the basin scale, mean annual P increased significantly in the Northwest (+20%), Southeast (+5%), and Songhuajiang (+2%) basins. No change in average precipitation was found in the Yangtze River as the increase in lower reach counterbalanced the decrease in the upper reach. The reductions in P in the other six basins varied from 2% to 7%. The relative changes in P and PET show that increasing P has been the major cause of the wetting trends in the

Southeast (29 wetting sites and 4 drying sites) and Northwest (89 wetting sites and 12 drying sites) basins. Particularly, wetting trends can be found in all four seasons in the arid Northwest driven by the increasing precipitation. The Hai River and Yellow River basins have become drier largely due to the reduction in summer precipitation, where a drying signal can be found at 67% (24 out of 36) and 73% (57 out of 78) of the sites in each basin. The changing PET also significantly altered the aridity. Decreases in basin-average PET occurred in all the basins except for the Yellow River and Southwest basins. The magnitude of PET reduction exceeded the decrease in precipitation in the Huai River basin, which was the major cause of wetting. It should be noted that there were large discrepancies in the energy condition across the climate stations within each basin. Particularly within large basins such as the Yangtze River and Yellow River basins, diverse climatic changes have triggered generally different wetting/drying patterns in the upper-reach and lower-reach regions.

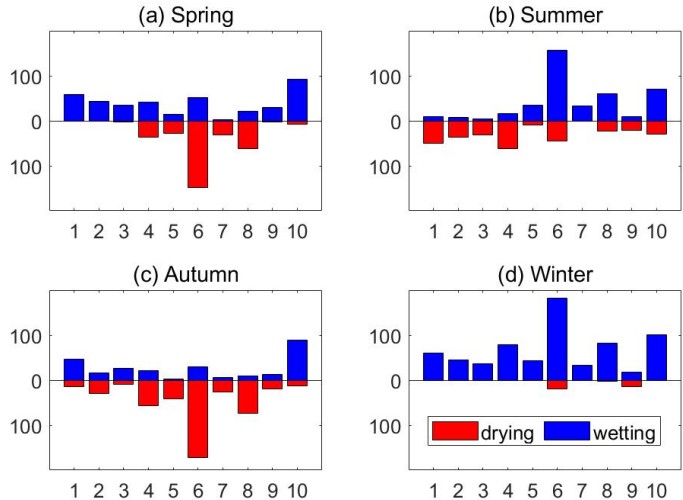

**Figure 4.** Number of drying/wetting climate stations in spring (**a**), summer (**b**), autumn (**c**), and winter (**d**) from 1961–1988 to 1989–2016 within the ten basins. On the x-axis, 1~10 represent the basins of Songhuajiang (1), Liao River (2), Hai River (3), Yellow River (4), Huai River (5), Yangtze River (6), Southeast (7), Pearl River (8), Southwest (9), and Northwest (10), respectively.

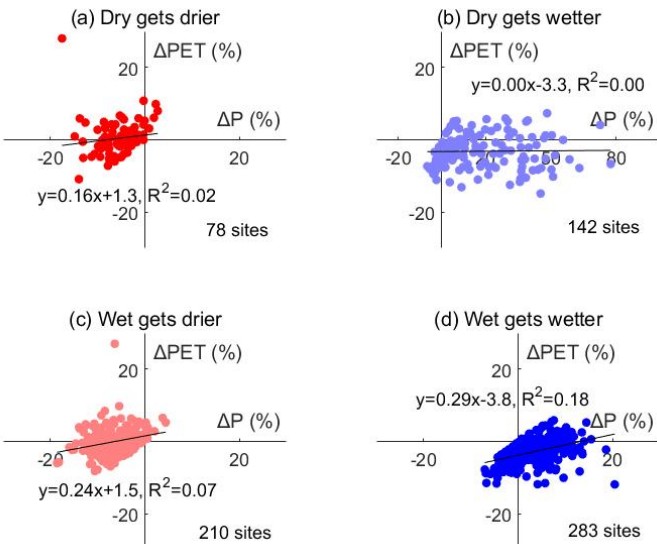

**Figure 5.** The "dry gets drier, wet gets wetter" pattern and changes in precipitation (ΔP) and potential evapotranspiration (ΔPET) at the 713 climate sites from 1961–1988 to 1989–2016. Each dot represents the results of a climate station. The climate sites were categorized into "dry" and "wet" when the aridity index (P/PET) in 1961–1988 was greater and less than 0.65, respectively.

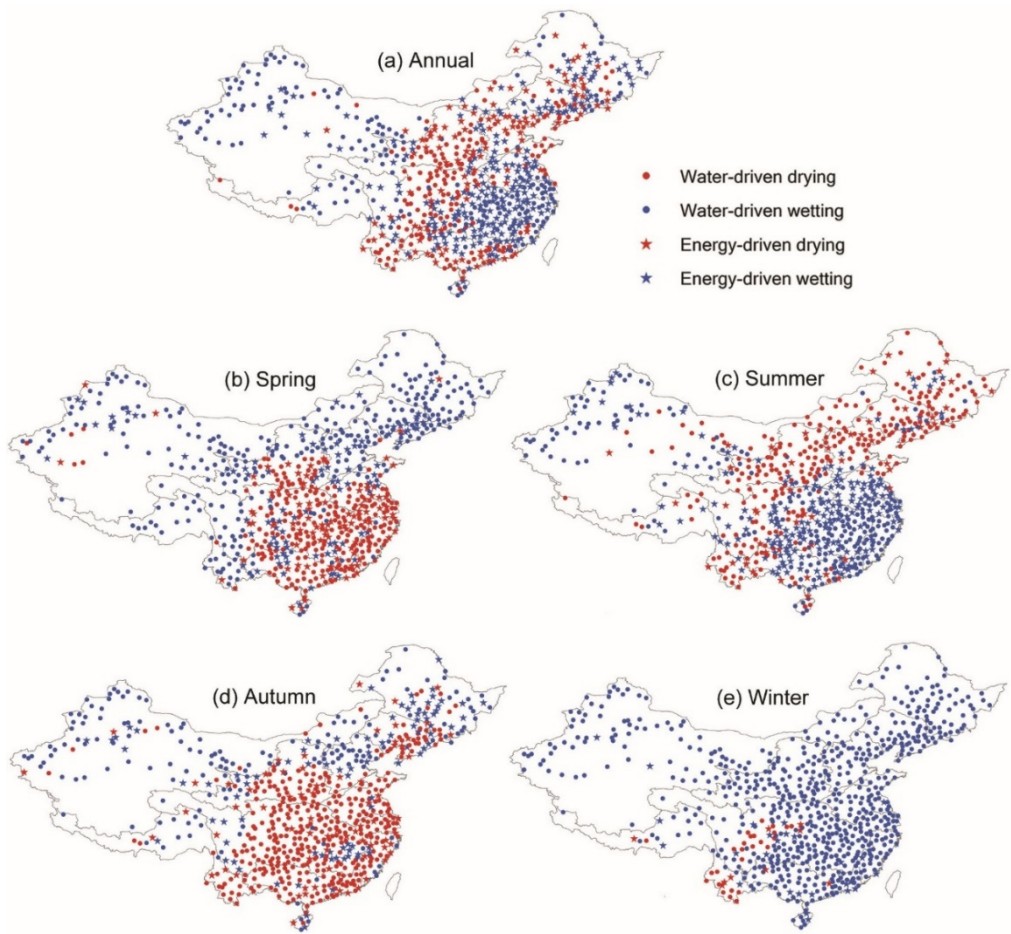

**Figure 6.** Driving factors of aridity changes at annual (**a**) and seasonal (**b**–**e**) scales across China. The drying/wetting site is marked as "water-driven" where precipitation was identified as the largest contributor to aridity change, and "energy-driven" where wind, radiation, or humidity was identified as the largest contributor.

### 3.3. Independent Effects of individual Climatic Variables

To further investigate the independent effect of each climatic variable, we attributed the changes in aridity from 1961–1988 to 1989–2016 to the changes in the six individual climatic variables (Figure 7). The independent effects of P at site level straddle wide ranges across the zero line within each basin. The distributions of median values and coverages of the 25% quartile to the 75% quartile suggest that positive effects of P on AI can be found at most sites in the Northwest and Southeast basins. In the meantime, negative effects of P prevailed within the Liao River, Hai River, and Yellow River basins. The most notable decrease in P occurred in the Hai River (−7%) and Yellow River (−4%) basins.

Unlike the diverse changes in P, the other climatic variables showed highly consistent changing directions at basin scale. While the increasing Tmax and Tmin (0.5 ± 1.5 °C) and decreasing Rh (−0.01 − −3%) enhanced PET and led to land surface drying, the reductions in Ws (−5 − −18%) and Rs (−0.4 − −7%) suppressed evaporation and drove the AI to increase. For example, the significantly increasing T and decreasing Rh caused the basin-average AI in the Hai River basin to decrease by −0.36 and −0.13, while the decreasing Ws and Rs drove AI to increase by +0.32 and +0.19 at the same time and offset the drying trend induced by T and Rh.

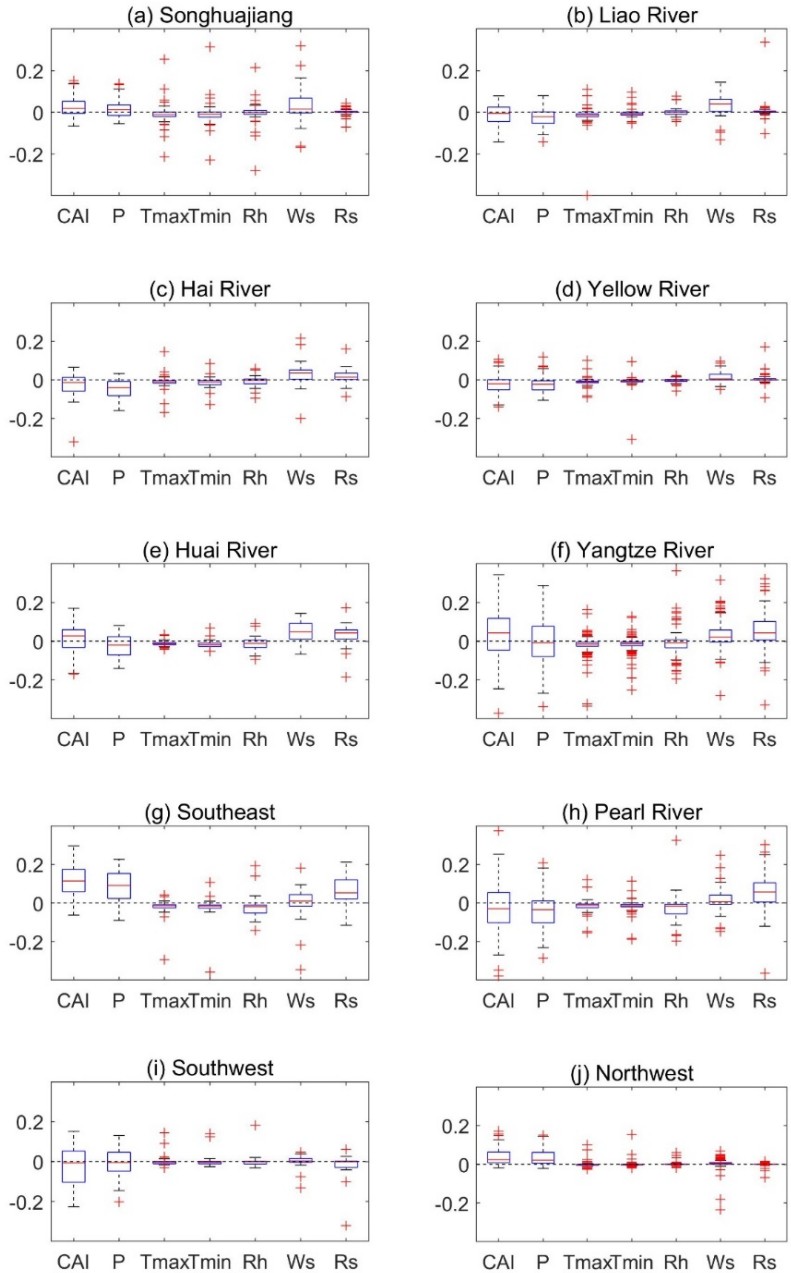

**Figure 7.** Attribution of changes in the aridity index (CAI) from 1961–1988 to 1989–2016 in the ten basins. The climatic variables are denoted by P (precipitation), Tmax (maximum temperature), Tmin (minimum temperature), Rh (relative humidity), Ws (wind speed), and Rs (radiation) on the x axis. The vertical spread of the box-whisker plots shows the variations in results among climate stations within each basin. The boxes cover the ranges from the 25% quartile to the 75% quartile of the distributions, with the median values marked by red lines within each box and outliers marked by plus signs.

Basin-average contributions of the climatic drivers (Figure 8) suggest that P was the largest contributor in most of the basins, with the relative contributions varying from 27% (Hai) to 58% (Northwest). The only exception was the Liao River basin, where changes in Ws, P, and T can explain 32%, 28%, and 26% of the AI change, respectively. Ws can also be recognized as the second most prominent factor in the Songhuajiang (29%), Hai River (25%), and Huai River (24%) basins. Meanwhile, changes in Rs played a major role in altering water-energy balance in the Southeast, Pearl, and Yangtze basins, accounting for 28%, 26%, and 23% of the changes in AI, respectively.

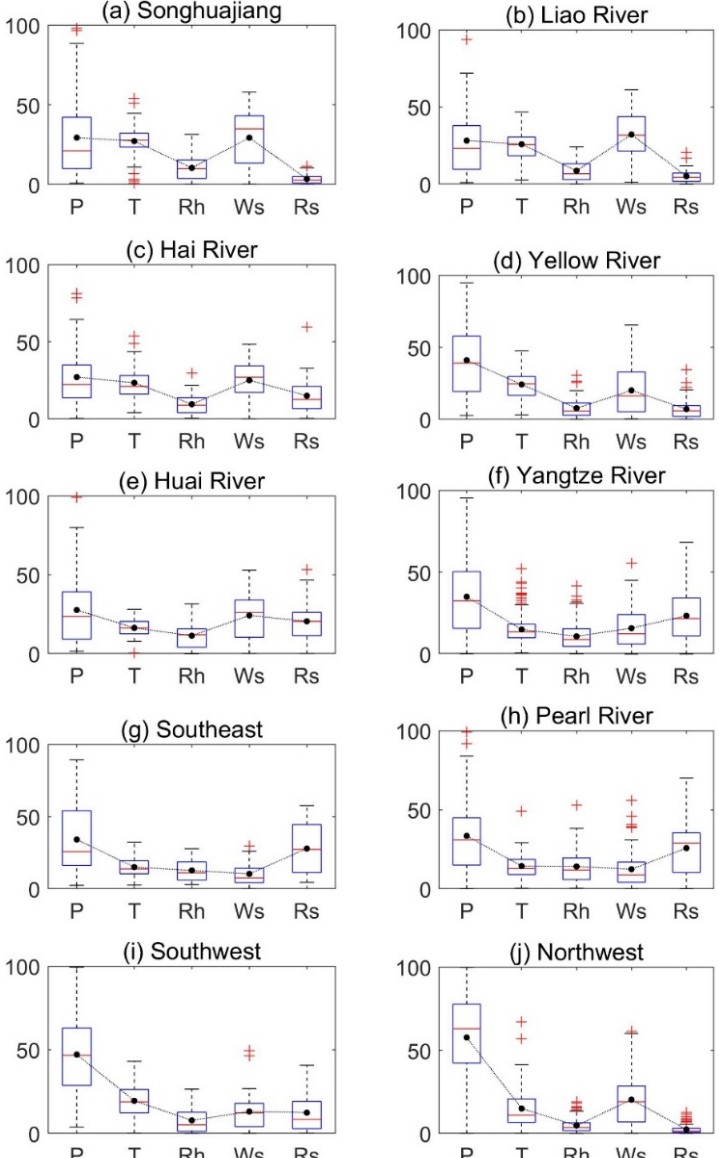

**Figure 8.** Relative contributions (%) of climatic drivers to the changes in aridity from 1961–1988 to 1989–2016 in the ten basins. The contributors are denoted by P (precipitation), Tmax (maximum temperature), Tmin (minimum temperature), Rh (relative humidity), Ws (wind speed), and Rs (radiation) on the x axis. The vertical spread of the box–whisker plots shows the variations in relative contribution among climate stations within each basin. The boxes cover the ranges from the 25% quartile to the 75% quartile of the distributions, with the median values marked by red lines within each box and outliers marked by plus signs. The mean contributions of each factor are marked with black circles.

The results of attribution analysis (Table 3) suggest that P was the largest driver of AI change at 51% (367 out of 713) of the climate sites, including 60% (174 out of 288) of the drying sites and 45% (193 out of 425) of the wetting sites. On the other hand, T was the most influential driver of PET change at only 28% (199) of the 713 sites, although the warming trend covered nearly the entire country (Figure 2). Ws, Rs, and Rh were the largest contributors to PET change at 37% (267), 27% (190), and 8% (57) of the sites, respectively. When compared to the individual effects of P, the sites where Ws, Rs, T, and Rh were identified as the largest contributors to AI change accounted for 22% (156), 17% (118), 8% (60), and 2% (12), respectively. It is clear that Ws and Rs were the second and third most influential climatic drivers of aridity change across the country, following P. We then cross-compared the drying/wetting patterns and the changing directions of the largest climatic drivers to examine

the consistency between them. The results show that the drying and wetting trends were consistent with the independent effects of the largest drivers at 93% of the sites, implying the dominant role of fast-changing individual variables at site level. Unsurprisingly, the decreasing Ws and Rs were found to be the dominant cause of wetting at 15% (109) and 13% (96) of the sites, respectively. The effects of decreasing Ws and Rs were overwhelmed by the combined effects of other variables at another 4% (28) and 2% (12) of the sites, where they were respectively identified as the largest individual contributors.

**Table 3.** Cross-comparison of the drying/wetting patterns and the changing directions of the largest climatic drivers from 1961–1988 to 1989–2016. The 713 climate stations are categorized into five groups according to the largest climatic driver of drying or wetting, including precipitation (P), temperature (T), relative humidity (Rh), wind speed (Ws), and radiation (Rs). Increase and decrease in the climatic drivers are denoted by "↗" and "↘", respectively.

| Largest Driver | Drying/Wetting Pattern | No. of Sites | Changing Direction | No. of Sites | Changing Direction + Drying/Wetting | No. of Sites |
|---|---|---|---|---|---|---|
| P | Dry gets drier<br>Wet gets drier | 5.3%<br>19% | P ↘ | 25% | P ↘ + drying | 24% |
| | Dry gets wetter<br>Wet gets wetter | 13%<br>14% | P ↗ | 27% | P ↗ + wetting | 27% |
| T | Dry gets drier<br>Wet gets drier | 2.2%<br>3.5% | T ↗ | 6.0% | T ↗ + drying | 5.8% |
| | Dry gets wetter<br>Wet gets wetter | 0.7%<br>2.0% | T ↘ | 2.4% | T ↘ + wetting | 2.4% |
| Rh | Dry gets drier<br>Wet gets drier | 0.0%<br>0.6% | Rh ↘ | 0.8% | Rh ↘ + drying | 0.6% |
| | Dry gets wetter<br>Wet gets wetter | 0.1%<br>1.0% | Rh ↗ | 0.8% | Rh ↗ + wetting | 0.8% |
| Ws | Dry gets drier<br>Wet gets drier | 3.2%<br>3.4% | Ws ↗ | 2.7% | Ws ↗ + drying | 2.7% |
| | Dry gets wetter<br>Wet gets wetter | 5.3%<br>10% | Ws ↘ | 19% | Ws ↘ + wetting | 15% |
| Rs | Dry gets drier<br>Wet gets drier | 0.1%<br>2.9% | Rs ↗ | 1.4% | Rs ↗ + drying | 1.4% |
| | Dry gets wetter<br>Wet gets wetter | 0.4%<br>13% | Rs ↘ | 15% | Rs ↘ + wetting | 13% |

## 4. Discussion

### 4.1. Linkage with Large-Scale Climate Change and Socio-Economic Developments

The observed changes in climatic variables across China are connected to the evolution in global climate systems and monsoon circulation [39]. As the East Asian monsoon weakens, monsoon precipitation intensifies in the south, but decreases in the Northern China Plain across the Hai River, Huai River, and Yellow River basins (Figure 6) [40]. Anomalies in moisture transport through the South Asian monsoon also affect precipitation in the Southwest and the upper reaches of Pearl River and Yangtze River basins. A weakened moisture supply from the Indian Ocean could be largely responsible for the drying trends over these areas [41].

Besides the background climate change, the variations in regional aridity and climatic drivers are intrinsically affected by local socio-economic developments. Over the past four decades, China has experienced both economic growth and environmental change at an unprecedented scale in human history [42]. Ecological restoration projects have been implemented since the late 1970s and accelerated after the 1980s as the financial support increased, and these have significantly enhanced vegetation growth. Satellite images suggest that leaf area in China increased by 1.35 million $km^2$ during 2000–2017

and accounted for 25% of the global net increase [33]. A plausible hypothesis for terrestrial stilling is an increase in surface roughness after revegetation in land surface [43]. The widely observed reduction in Ws over northern China could be partially due to the implementation of the "Great Green Wall" project, where vast forests were planted across the northern aridlands of China to reduce desertification [44]. The increasing coverage of forests and croplands could also enhance local air humidity when soil moisture is sufficient and lead to a lower evaporative demand and higher AI.

Meanwhile, rapid urbanization, particularly after 1980s, has caused profound climatological effects across China. Urbanization causes landscape fragmentation and alters local weather patterns such as the processes of cloud formation, turbulence, and convection [45]. The widely observed increases in T and decreases in Ws and Rh have stimulated studies on the climate–land surface interactions in urbanized areas, such as the well-known urban heat island effect [46] and urban dry island effect [47]. Moreover, large amounts of anthropogenic aerosols have been released into the atmosphere along with urbanization and industrialization. These air pollutants can enhance the scattering and absorption of solar radiation, disturb the formation of precipitation, and alter regional-scale water and energy balances [48,49]. Although the impacts of aerosols on local climate vary with atmospheric backgrounds and cloud types, large-scale air pollution is likely to suppress eco-hydrological fluxes and compromise water yield and ecosystem productivity [50]. Previous climate modeling studies also suggest that increasing anthropogenic sulfate aerosol emissions over the Northern Hemisphere could be the dominant cause of Asian summer monsoon weakening [51].

Generally speaking, energy-driven changes in aridity were mostly observed throughout eastern China, and the impacts of changes in wind and radiation were more prominent in the north (i.e., the Liao, Songhuajiang, Hai, and Huai River basins) and south (i.e., the Southeast, Pearl, and Yangtze basins), respectively. Although there is increasing evidence that these changes in climatic variables are associated with the growing population and great anthropogenic disturbances in the past several decades [21,43,52], further studies are needed to identify how these disturbances contribute to the changes in specific variables and the subsequent land surface drying and wetting.

### 4.2. Implications for Water and Land Management

Our results have important implications for water and land management in a changing climate across China. Water resources planning needs to prepare different management strategies for regions facing contrasting hydro-climatological conditions. Additional water storage and regulation facilities are needed in wet-gets-wetter regions such as the lower reach of the Yangtze River and Southeast basins. Precipitation-induced wetting and warming-induced changes in snowmelt runoff can also cause more flash floods in the arid Northwest [53]. On the other hand, the decreasing rainfall results in the tendency toward increased droughts in northern China. The most notable reduction in precipitation occurred in the Hai River (−7%) and Yellow River (−4%) basins. These two basins are densely populated arid regions and have been the most water stressed areas in the past decades. The situation could be worsening as climate change and anthropogenic disturbances intensify [54]. As well as the ongoing South-to-North water transfer project [55], other water conservation measures such as upgrading the economic structure, harvesting storm rainfall, promoting waste water recycling technology, and improving water use efficiency should be implemented to alleviate water stress in these regions [56,57].

The vast croplands across northern China (i.e., the Songhuajiang, Liao, Hai, Yellow, and Huai River basins) are threatened by rising temperatures and diminishing water availability for irrigation and food production, particularly the drying trends in summer (Figure 6c) when paddy rice fields require large amounts of water. Adaptations in the agricultural system and irrigation technology are needed to secure crop supply and to reduce vulnerability to climate change and droughts [58,59]. The warmer and drier conditions may also result in higher risks of insects, disease, and catastrophic wildfires in forests and grasslands [60,61]. Proper land management practices such as forest thinning and planting drought-tolerant species are critical to increase resiliency to climate change and reduce the potential threats to ecosystems and society [62,63]. Land management policies need to consider

the drying/wetting trends of the land surface and variations in the regional water demands of human and ecosystems. For example, the wetting and warming trends have facilitated revegetation efforts in western China, while the increasing water uses for revegetation (i.e., evapotranspiration) and the drying climate have caused significant decline in runoff in the Yellow River basin and have challenged the sustainability of the water resources system [64]. It is vital to consider the trade-offs among ecosystem services, water security, and economic growth, and to balance the environmental and socio-economic demands in a changing climate.

### 4.3. Caveats

Several limitations and caveats apply to our study. Considerable uncertainties could be involved in the long-term climate data due to the changes in instruments, data management, and surroundings of climate stations during the past decades. In particular, humidity, wind, and radiation components have not been as widely observed and analyzed as precipitation and temperature. These uncertainties are unlikely to challenge our findings at the basin scale, yet could alter the site-level quantifications of individual variables' effects to various extents. Moreover, we have focused on attributing the observed changes in PET and AI to individual climatic variables based on the Penman–Monteith equation in this assessment. We did not consider the interactions among these variables. For example, changes in local moisture supply and radiative and aerodynamic conditions also affect the processes of precipitation formation, and thus lead to changes in aridity indirectly [49,50]. The relative roles of precipitation and other variables could be different when causality is interpreted from different temporal scales due to the complex interactions among aerosols, greenhouse gases, and cloud properties [65]. More efforts in on-site weather monitoring, data assimilation, and interdisciplinary research are needed to better understand the relationship among varying individual climatic factors and climatic responses to human activities.

## 5. Conclusions

This study examined the independent contributions of individual climatic drivers of aridity change across China, and particularly highlighted the roles of terrestrial stilling and solar dimming in a warming climate. Results suggest that the drying and wetting features varied greatly across climatic regimes and river basins during the past six decades. Significant wetting can be mostly found in the Northwest, Songhuajiang, lower Yangtze River, and Southeast basins, while significant drying occurred extensively in the Southwest, Yellow River, upper Yangtze River, and upper Pearl River basins. Inhomogeneous drying/wetting paradigms were also detected across the seasons. In spring and autumn, it became more humid in western and northeastern China, but more arid to the south of the Huai River basin (i.e., Huai, Yangtze, Southeast, and Pearl). On the contrary, summer precipitation was enhanced in the south yet suppressed in the north as the East Asian monsoon weakened, causing wetting trends in the lower Yangtze and Southeast and drying trends in the Hai and Yellow River basins. A consistent wetting signal in winter, which was mostly driven by the increasing precipitation, was found over the entire country.

While changes in precipitation differed spatially, increases in both maximum and minimum daily temperature were observed at nearly all of the climate stations. The warming and reductions in relative humidity significantly enhanced PET and caused the land surface to be drier across China. On the other hand, significant decreases in wind speed and radiation affected more than two thirds of the sites and caused widespread decreases in PET. Such trends of terrestrial stilling and solar dimming largely offset warming-induced drying and were found to be the dominant causes of wetting at 15% (109) and 13% (96) of the sites, respectively. In particular, the role of decreasing wind speed was more prominent in northern China, and was identified as the largest contributor in the Liao River basin (32%) and the second largest in the Songhuajiang (29%), Hai River (25%), and Huai River (24%) basins. Meanwhile, reduction in radiation can explain a major part of AI changes in the south at basin level, including the Southeast (28%), Pearl River (26%), and Yangtze River (23%) basins.

Besides the impacts of weakening East Asian monsoon and South Asian monsoon, these changes in aridity and climatic drivers could be related to the rapid urbanization, anthropogenic aerosol emissions, and large-scale revegetation projects across China in recent decades. Our results highlight the need for more emphasis to be placed on the climatological impacts of regional-level anthropogenic disturbances under the background of climate change. Further studies on the interactions among individual climatic variables and multiple environmental stressors are warranted for sustainable water and land management.

**Author Contributions:** Conceptualization, K.D. and J.G.; methodology, K.D.; validation, T.H., Y.M., and X.W.; formal analysis, K.D.; investigation, K.D.; data curation, K.D.; writing—original draft preparation, K.D.; writing—review and editing, J.G.; T.H., Y.M., and X.W. All authors have read and agreed to the published version of the manuscript.

**Funding:** This work was supported by the Open Research Fund Program of State Key Laboratory of Water Resources and Hydropower Engineering Science (2017SWG01), the National Natural Science Foundation of China (51909285, 91647204), the Guangdong Provincial Department of Science and Technology (2019ZT08G090), the Yichang Natural Science Research Project (A20-3-005), and the Open Funding of Engineering Research Center of Eco-Environment in Three Gorges Reservoir Region, Ministry of Education, China Three Gorges University (KF2019-12).

**Conflicts of Interest:** The authors declare no conflict of interest.

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
