# Peer review of "Assessing the Roles of Terrestrial Stilling and Solar Dimming in Land Surface Drying/Wetting across China"

_water, doi:10.3390/w12071996_

Round 1
Reviewer 1 Report
Manuscript Ref: water-838353
Title: Assessing the roles of terrestrial stilling and solar dimming in land surface drying/wetting across China
Authors: Kai Duan, Jiali Guo, Tiesong Hu, Xianxun Wang, Yadong Mei
Journal: Water
The manuscript entitled “Assessing the roles of terrestrial stilling and solar dimming in land surface drying/wetting across China (ref water-838353)” focuses on the analysis of land surface drying/wetting trends over China using the data from 713 observational stations in the period 1961-2016. To do this, the author analyzed trends in the aridity index, but also the change in those variables involved in the drying/wetting paradigm (i.e., precipitation and potential evapotranspiration, the latter though changes in the maximum and minimum temperatures, relative humidity, and radiation).
The subject of this paper is of great interest to the study area to better understand the effect of the different variables involved in drying/wetting trends, so merits publication in Water journal. However, some aspects must be addressed before its publication.
General comments:
One of my major concerns is related to the title of the study. In my opinion, the title does not reflect the main aims of the paper. In this study, the authors analyzed (1) drying/wetting trends and (2) quantified the direct and compound effects of changing precipitation, temperature, wind, humidity, and solar radiation on aridity as indicated on lines 89-92. Therefore, the study is not centered on analyzing the role of the radiation and wind speed on the drying/wetting trends specifically. Also, the authors highlighted the terrestrial stilling and solar dimming concepts in the keywords, and, though the results seem to show an effect of wind and radiation on the drying/wetting trends, I think that these concepts are not worth mentioning based on the results as keywords. If the authors want to highlight the effects of the terrestrial stilling and solar dimming in aridity, they must make further analyses in this regard. Moreover, the results indicate the precipitation as the most important factor contributing to changes in aridity, so the title seems not to be appropriate either according to the conclusions of the study.
Concerning the aim (3) “to explore the linkages between the changing climate and large-scale anthropogenic disturbances and the implications for hydrological cycle across the major river basins” (lines 92-94). I am not sure that the authors have explored such linkages in the paper. In this regard, the authors indicate in their abstract that the changes due to radiation and wind speed could be associated with the wakening monsoon and anthropogenic disturbances (lines 33-36) but no analysis seems to be performed in this regard.
On the other hand, and for the other results, I think that more details and comments are needed in the results section. Sometimes, this section is difficult to follow and to add more comments could improve the reading of the manuscript.
Specific comments:
Section 3.4 (Independent effects of individual climatic variables): This section is one of the most interesting parts of the study. However, for me, it is difficult to follow the conclusions obtained by the authors in this section, especially in relation to Figure 7 and Table 2. I think that more details are needed. Also, I am not sure if the comments from line 238 to line 243 are related to Figure 7 or not (e.g., Figures 2 or 3, maybe?). I suggest rewriting the paragraphs from line 238 to line 255 in order to make more clear the reading of the manuscript. Also, I think that it’s important to add some sentences at the beginning of the description of figure 7 detailing how to interpret such a figure, and what information it provides.
Lines 264-280: I think that the comments related to Table 2 are difficult to follow. What is the purpose of this Table? Is it a summarize table? I would suggest more details about this and to briefly explain the terms added to the table.
Section 3.5 (lines 282-320): Are these results supported by any analysis? If not, I would move the section 3.5 to the Discussion section.
About Figures
Figure 2: I would change the caption of this figure. In my opinion the paragraph: “The increases and decreases in the variables that are positively correlated with AI (i.e., P, Rh, and AI) are denoted by blue and red dots, respectively. Contrarily, the increases and decreases in the variables that are negatively correlated with AI (i.e., Tmax, Tmin, Ws, Rs, and PET) are denoted by red and blue dots, respectively” (lines) is confusing.
Figures 3b and 4: I would add a short definition of “proportions of drying/wetting sites” at least in the caption of figure 3.
Figure 5: What is the meaning of “dry gets drier, wet gets wetter” pattern? How was it computed? What do these results represent?: I would include an additional description in the text about how was carried out the analysis in this figure. In this sense, I think that it could be good to remember the meaning of the “dry gets drier and wet gets wetter” paradigm in order to make the reading more clearly. Also, the results seem to show some relationship between precipitation and evapotranspiration, at least for three of the four scenarios (dry gets drier, wet gets drier, and wet gets wetter), so any adjust of the values (e.g., linear regression) and its r² value could be useful to comment the results for the figure. I also think adding the number of points for each scenario in the figure (e.g., the percentage of the total stations representing each scenario) could be good.
Figure 6: I would change the marks in Figure 6. Using different colors instead of different marks (i.e., stars and dots) could help to show the results more clearly. There are some regions (e.g., Southeast or Pearl River) where there are many stations, and therefore, stars and dots are not distinguished well.
Reviewer 2 Report
The paper is written at a good level and has significant results. However, there are some minor points:
L 116. The percentage of lags of more than one week of the total amount of data must be indicated.
L 126. Indicate the uncertainties in the values of as and bs.
Reviewer 3 Report
Dear authors,
Congratulations on an excellent, succinct paper. It flows and reads well, in general, and I have only minor comments. First, general points:
- There is too much filler in some longer sentences. Try to reduce the word count by removing non-essential parts of sentences
- More citations are needed throughout when introducing specific terms as "pan evaporation paradox", and/or a brief explanation of the term and its relevance.
- Symbols (Rh, Rs, etc) should be in a maths font so it isn't confused with English text.
Specific comments:
- Line 47 and throughout: "water balance components" should be "water-balance components", as for all compound adjectives (else it can be confusing to read, as here).
- Line 51: please clarify why precipitation would increase, but humidity goes down, when high (relative) humidity is a prerequisite for surface rainfall. Also, the use of "humidity" here and throughout (e.g., Line 54) is ambiguous. Please specify upon the first use of "humidity" whether it is relative, absolute, specific, etc.
- Line 55-56: consider removing "A rising concern in hydro-meteorological communities is that" and starting the sentence with "Many". This applies throughout: there are sentences that can be shortened by removing non-essential parts like this.
- Line 56: "Many climate(-)change studies..." needs a citation, for instance, a review paper. It is not clear if the following citations (15,16) are about papers addressing concerns, papers on studies with errors in their methods, or papers on the methods themselves.
- Line 57: change "temperature rising" to "rising temperature"
- Line 69: "...in the east." - east of where? USA?
- Line 78 and throughout: be consistent with your Oxford comma (before the final 'and' of a list). For instance, "...wide range of climatic, geophysical and ecological circumstances" does not have a comma as written, but Line 91 has an example with the Oxford comma. Both are fine (if MDPI house style allows it), but pick one for consistency.
- Line 81: "regions" - of China? Be specific.
- Line 85: "pan evaporation paradox" needs explaining and citing
- Line 99-100: What does "improved" specifically mean? Also needs a citation
- Line 103: "quality controlled" - how, specifically? Citation needed too.
- Line 130-131: you can remove the "According to the...Desertification" and just cite the book (consider adding the page, if that is possible with MDPI citation style).
- Line 131-133: consider putting these definitions into a table for the reader's ease. Then you can refer to the table throughout instead of reminding the reader constantly of definitions (see below)
- Line 165: remove "had".
- Line 178: should "threshed" be "threshold"?
- Line 179: "In the northeastern..." should be "In northeastern..."
- Line 181 and throughout: in this section, you often state the definition (arid) and the definition ("as the average AI fell below 0.65"), which defeats the purpose of having the categories as shorthand in the first place. If you put these definitions in a table (as suggested above), you can remind the reader once more that the definitions are in the table.
- Line 188: "occured" should be "occurred"
- Line 199: "accoundted" should be "accounted"
- Line 215: "Wetting signal were" should be "signals were" or "signal was"
- Line 223: clarify if the interdependence of precipitation and humidity is a problem for your analysis?
- Line 226-227: "wetting trend can" should be "wetting trends can"
- Line 227: "all the four" should be "all four"
- Line 253 and throughout: you say "caused" here and elsewhere. Can you claim causation?
- Line 292: The unprecedented growth needs a citation
- Line 298: "Great Green Wall" needs a citation and a brief explanation of what it is
- Line 302: need to cite why urbanisation alters weather patterns (or you could word this better as "alters flow in the planetary boundary layer").
- Line 305-306: cite both dry-island and heat-island effects (not just one)
- Line 306-307: citation needed for aerosol effect
- Line 332: "South-to-North Water Transfer" is in title case, but it needs a brief definition, and it needs a citation.
- Line 367: It is good you have a paragraph addressing your limitations. But consider adding a line defending your work in the face of these caveats, and how they might change the results, and why you think your findings are robust enough to publish despite that.
